# Toward developing a sustainability index for the Islamic Social Finance program: An empirical investigation

Tika Widiastuti[1], Arie Prasetyo[1], Anidah Robani [2]*, Imron Mawardi[1], Rida Rosida[3], Muhammad Ubaidillah Al Mustofa[4]

1 Department of Sharia Economic, Faculty of Economic and Business, Universitas Airlangga, Surabaya, Indonesia, 2 Institute of Technology Management and Entrepreneurship, Universiti Teknikal Malaysia Melaka (UTeM), Melaka, Malaysia, 3 Department of Islamic Economic and Finance, Universitas Pendidikan Indonesia, Bandung, Indonesia, 4 Department of Development Studies, Institute Teknologi Sepuluh Nopember, Surabaya, Indonesia

* anidah@utem.edu.my

**Data Availability Statement:** Supporting information files are deposited at Qualitative Data Repository with doi:10.5064/F6TLONN4.

## Abstract

Several previous studies state that the Islamic Social Finance program has not fully succeeded in creating prosperity, and there are no definite measurements to show the sustainability impact of the program. Thus, a measurement is needed to analyze various aspects in achieving the success and sustainability of Islamic social finance programs. This study developed an index for performance evaluation with an emphasis on the success and sustainability of the Islamic Social Finance program. The study used the Analytical Network Process to determine and analyze priority components. Furthermore, the Multistage Weighted Index method was used to calculate the final index score. The index was built by taking into consideration various factors, stakeholders, aspects, and indicators. This study indicates that aspects of funding contribution from donors (0.22), involvement of donors in giving advice (0.99), and controlling of supervisor (0.08) are priority aspects in the success and sustainability of the program. An empirical investigation was performed on three different programs in Indonesia: A, B, and C. Program A (0.81) and C (0.80) have succeeded in improving the beneficiaries' quality of life to the level of economic resilience, although at a low level of sustainability (76–84.33). On the other hand, program B (0.73) is at the economic reinforcement and has not yet achieved sustainability. This index can be seen as a comprehensive tool for measuring the success and sustainability of the program at several levels.

## Introduction

The management of Islamic Social Finance continues to grow, including the distribution of funds [1–3]. Initially, social finance management channeled Islamic Social Funds in consumptive forms. Also, the waqf fund is still generally distributed for non-productive forms. However, in line with the goal of Islamic Social Finance in achieving prosperity, the Islamic social

**Funding:** This is a funded studies. It is funded by Universitas Airlangga (UNAIR), East Java, Indonesia. But the funders play no role in the study design, data collection and analysis, decision to publish or even the preparation of manuscript.

**Competing interests:** No authors have competing interests.

funds began to be channeled into a more productive state. One of the effective distributions is through business capital provision, training, and mentoring [4–6]. Effective forms of distributed funds develop knowledge and skills and impact beneficiaries' income increase and independence [1, 2].

As a Muslim majority country, Indonesia has a great potential for Islamic Social Finance instruments. According to the recent reported data, the potential for Islamic Social Finance instruments have reached USD 22.8 billion for zakat [7], USD 137.9 million for land waqf [8], and USD 13 billion for cash waqf. However, Islamic social funds channeled to official Institutions only reached USD 31 million. This figure is considered insignificant because the distribution of Islamic social funds outside the institution/non-administrative reached IDR 30 trillion [7]. With this fact, Islamic social funds that are professionally managed and distributed through effective empowerment programs are limited. For this reason, Islamic Social Finance must prepare and implement the empowerment program mindfully to have a positive and significant impact on the quality of life of the beneficiaries [5, 6].

Indonesia is the most generous country globally [9, 10]. In contrast, society does not yet fully have the literacy and belief that Islamic social funds must be managed professionally through the Institution [11–14]. This condition is not ideal because people who channel their funds directly to the community are generally in the form of consumptive distributions that do not have a long-term effect on improving the economic, social, spiritual status, and abilities of beneficiaries.

The urgency of Islamic social funds professional management is reflected through empowerment programs. First, distributing Islamic social funds can achieve welfare by maintaining five aspects of maqashid sharia [15] so that the community can sustain the quality of human life. Second, Indonesia's poverty rate is 144 out of 172 countries in the world and 6 out of 10 countries in Southeast Asia [16]. This ranking is higher than Malaysia, a zakat collecting and Muslim-majority country, ranking 8. Poverty in Indonesia reached 10.4%, exacerbated by the poverty depth index, which is 1.71 points. The higher the poverty depth index value, the farther the average public expenditure is from the poverty line [17]. Poverty is also a severe problem in other Muslim-majority countries, such as Pakistan, Afghanistan, and Bangladesh [16]. These countries are also concerned about managing Islamic social funds. Third, the Gini ratio problem is enormous. Based on the data from Statistic Indonesia [17], Indonesia's inequality level in March 2021 reached 0.384, an increase of 0.003 from March 2020. Fourth, Indonesia's Human Development Index (HDI) level only grew by 0.30% in 2020 [18] The inequality level needs to be of concern, where the HDI reflects that the population has difficulty obtaining income, education, health, and other development outcomes. Fifth, several prior studies show that Islamic social funds have not fully impacted welfare [19–21]. For these reasons, the community must manage Islamic Social Finance professionally through potential empowerment programs.

The management of the Islamic Social Finance program is closely related to sustainability [22, 23], where Islamic Social Finance plays a role in ensuring the sustainability of the beneficiaries' life [24, 25]. Therefore, the Islamic Social Finance program is not only directed at the success of improving the quality of life of beneficiaries so that they can get out of the poverty but also at the sustainable quality of life. Conceptually, in Brau et al. [26], there are two paradigms in sustainability, namely, institutionalist and welfarist paradigms. The institutionalist paradigm considers that sustainability is closely related to the ability of institutions to obtain funds so that they can be independent and not dependent on others. In contrast, the welfarist paradigm considers the main point: the institutions can feel the benefits of sustainability.

In the context of the Islamic Social Finance program, by referring to the abovementioned definition of sustainability, it can be concluded that sustainability occurs when (1) the program

can encourage the beneficiaries to be independent (able to meet their own needs) and (2) the program can prosper the beneficiaries. Thus, it needs various inputs and appropriate indicators to guide the empowerment programs and measure their success and sustainability [22]. Moreover, empowered programs in Indonesia are diverse due to various Islamic Social Finance institutions in the country [27].

Several studies have developed Islamic social fund indexes/indicators based on the literature review, focusing on institutional and program indexes. Amin [28] developed an index for the impact of zakat on Muslimpreneurs. On the other hand, Abdullah et al. [29] developed the Zakat Index (ZEIN) to measure the effectiveness of the Zakat Institution. Further, Wahab et al. [30] built a service quality index for Zakat Institutions. BAZNAS Center Strategic Studies, a government institution that focuses on zakat research, has developed several indices, including Zakat Utilization Index, Zakat Village Index, Zakat Transparency Index, National Zakat Index, BAZNAS Welfare Index, Zakat Coordination Index, Zakat Literacy Index, and Zakatnomics Development Index. In addition, the National Waqf Board developed National Waqf Index. Previous research has focused on measuring institutions and programs from just one aspect, such as service quality [30] and accountability. The index in this study focuses on the concept of sustainability and maqashid sharia, based on the role of all stakeholders in the management of Islamic Social Finance.

In contrast to previous research, this study developed a new performance measurement model for the Islamic Social Finance program. This research has several novelties. First, this study develops previous research by establishing an index to see the success and sustainability of Islamic Social Finance programs. Second, the researchers conducted empirical investigations on three Islamic Social Finance programs to measure their success and sustainability. Third, the main criterion of the empowerment program is to achieve prosperity for beneficiaries, hence, this research also emphasizes the importance of integrating Islamic social funds in achieving prosperity. Finally, this research built a comprehensive measurement, in which researchers can assess a program for its level of success in transforming the welfare of beneficiaries, represented through 4 ER (Economic Rescue, Economic Recovery, Economic Reinforcement, Economic Resilience).

This study indicates that the funding contribution from donors (0.22), some advice from donors (0.9), and control by supervisor (0.8) are priority aspects in achieving the success and sustainability of the program. An empirical investigation conducted on three programs—A, B, and C—shows that programs A (0.81) and C (0.80) are at the level of economic resilience but with low sustainability. Meanwhile, program B (0.73) has not yet achieved sustainability and is still at the level of economic reinforcement.

This research has implications in several ways. First, this study emphasizes the importance of the involvement of donors in paying Islamic social funds to institutions to be managed productively. Islamic Social Finance institutions must formulate strategies to increase donor engagement, such as providing various payment platforms that make it easy or creating potential programs that have long-term impacts. The government, as a regulator, can issue a series of policies and regulations that advise the public to pay Islamic social funds through Institutions. Second, increasing public literacy related to Islamic Social Finance is essential to increase donor involvement. Third, an empirical investigation of the Islamic Social Finance Program at the level of economic reinforcement and economic resilience with low sustainability shows that several aspects need to be improved. As program managers, Islamic Social Finance Institutions must formulate various program optimization strategies and coordinate with multiple stakeholders, such as regulators, academics, associations, and supervisors.

As for the structure of this paper, this article is divided into five main sections. The first section discusses the background of the writing. The second section presents the theoretical basis

of index development. The third section describes the methods used. The fourth section identifies the results and analysis. Finally, the conclusions are shown in the fifth section.

## Literature review

### Islamic social finance concept

Islamic Social Finance is an integrated socio-economic empowerment system divided into traditional and contemporary financial systems. Traditional Islamic Social Finance instruments classified into two types: Philanthropy-based instruments, such as zakat, infaq alms, and waqf, and cooperation-based instruments, such as qardh and kafalah [31]. Meanwhile, contemporary Islamic Social Finance instruments are in the form of Islamic microfinance. Philanthropy has different shapes and definitions [32]. This concept involves four main parties: social welfare initiators, social finance providers, social ecosystem coordinators, and beneficiaries [33]. Each of these parties plays a clear role in the conception, implementation, coordination, financing, and improvement [25]. Some of the expected goals of Islamic social finance are alleviating unemployment and poverty [34], considering the ethical orientation created between awareness and knowledge of the programs of Islamic Social Finance [35].

The previous research shows differences in understanding Islamic Social Finance (ISF) instruments (see Table 1). Zakat has several characteristics that distinguish it from other types of levies, including (1) as a religious obligation where the subject includes Muslims or business entities (corporations or companies) owned by Muslims, (2) organized manner of zakat collection managed by the government, (3) the object of zakat is widely imposed on all business activities ranging from livestock, agriculture, commercial activities, mining, to immovable assets, (4) imposing on individuals with the nishab (limit) or minimum wealth, and (5) the recipients of zakat (mustahiq) have been determined in the Qur'an, in the form of eight groups.

On the other hand, waqf is a voluntary donation generally also known as part of sadaqah (good deeds) and infaq (expenditures to please Allah) [42]. Islam makes waqf one of the divine gifts for society's prosperity (religion bounds the community) [43]. The primary difference from the waqf instrument is that the donation should not reduce the principal value of the waqf property. Consequently, the waqf property management must put Sharia compliance as a priority. Kayikci [44] interprets infaq as a gift to Allah and an investment of eternal goodness. Infaq in the Islamic system conceptually implies giving, including the giver and the relatives, for a better society and its people [45]. In infaq, there are no provisions regarding the form, implementation time, or the amount. Also, sadaqah is a type of good deed broader than zakat

**Table 1. Instruments of Islamic Social Finance.**

| No | Prior Research | Islamic Social Finance Instruments | | | | | | | |
|----|----------------|------|-------|-------|------|-------|------|------------|--------------|
| | | **Waqf** | **Zakat** | *Infaq* | **Alms** | **Grant** | **Qard** | **Qard Hasan** | **Microfinance** |
| 1 | IRTI [31] | √ | √ | √ | √ | √ | √ | √ | √ |
| 2 | Jouti [25] | √ | √ | | √ | | √ | | √ |
| 3 | Usman et al. [36] | | √ | √ | √ | | | | |
| 4 | Iskandar et al. [37] | √ | √ | √ | √ | | | | |
| 5 | Ascarya [38] | √ | √ | √ | | | | | |
| 6 | Shuaib and Sohail [39] | √ | √ | | √ | | | | |
| 7 | Khan et al. [40] | √ | √ | | √ | | | | |
| 8 | Tok et al. [41] | √ | √ | | √ | | | | √ |

Sources: Authors' Compilation (2022)

and infaq. Qard Hasan is one of the tools to achieve economic and social justice as aspired by Islamic economics [46]. With Zakat, Qard Hasan can be used for financial inclusion because the poor can have easier access to financial services at lower costs if Islamic Financial Institutions can channel them effectively [47]. Referring to the explanation above, what is meant by ISF is zakat, infaq, alms, and waqf. Furthermore, this study will examine the integration of zakat, infaq, alms, and waqf funds and analyze how they influence the welfare of mustahiq.

## Concept of sustainability in Islamic Social Finance programs

In the context of Islamic Social Finance, several studies have discussed sustainability and Islamic Social Finance. Khalfan et al. [48] stated that there are three systems in waqf sustainability: beneficiaries, property, and maintenance. Robani and Kamal [49] researched the potential of Islamic Social Finance for sustainable development, focusing on social justice, balance, and how it produces maslahah for the people. Sulaiman and Alhaji [50] analyzed the financial sustainability of waqf institutions. They used the Tuckman and Chang [51] model to measure sustainability from four aspects: operational margin ratio, equity balance, revenue concentration, and administrative cost. Jouti et al. [25] built a sustainable Islamic Social Finance ecosystem with an integrated approach, which indicates the need for cooperation of all stakeholders -zakat institutions, waqf institutions, banking, Islamic microfinance institutions, Islamic capital markets, etc.- in creating a sustainable ecosystem. Hassan et al. [52] explained that sustainability in waqf includes three elements: maintenance of the waqf property, increasing the productivity of waqf, and increasing the chance of providing benefits to the community. Sapuan and Zeni [53] described that efficient management is the key to achieving the sustainability of Islamic Social Finance. This study examined the sustainability of Islamic Social Finance in terms of its institutions and instruments.

The program and how it can create sustainability for the beneficiaries also mirror the sustainability of Islamic Social Finance. One of the critical goals of Islamic Social Finance programs is to ensure that sustainability is the empowerment program. However, research on empowerment programs still focuses on impact measurement [6, 14, 54–57]. Others indicate that the empowerment program had not yet fully impacted beneficiaries [19, 20, 58]. On the other hand, research related to Islamic Social Finance and sustainability programs is still limited. Hence, this study attempted to build a comprehensive index of the success and sustainability of Islamic Social Finance Programs.

Based on Brau et al. [26], there are two paradigms in sustainability: Institutionalist and welfarist. Institutionalist refers to financial self-sufficiency, where the provision of services for the poor depends on the institution's sustainability closely related to the institution's ability to obtain funds continuously. Welfarist emphasizes that sustainability is closely related to poverty alleviation, where sustainability is assessed from social motives, not profit motives.

Pischke [59] analyzed the sustainability of microenterprises of lenders and stated three sustainability levels. Level 1 of sustainability is a healthy portfolio flow where there must be a balance between outflows, inflows, and generated surplus. In achieving this level, the enterprise needs excellent risk management. At level 2, the enterprise is considered sustainable if the generated surplus at level 1 can cover operational costs. The third level of sustainability is how the generated surplus can cover other expenses after being used to cover operating costs. According to Morduch [60], who analyzed the sustainability of microfinance schism, sustainability refers to returning the employed capital and obtaining sufficient profits to increase the invested capital. The existence of products, services, management practices, risk management, targeting, supporting policies, regulations, and impact assessments is essential to achieving sustainability [26, 59–61].

Based on the research above, it can be concluded that the sustainability parameters of the Islamic Social Finance program are: (a) it creates financial self-sufficiency for the beneficiaries where they become financially independent, (b) it supports the recipients to generate income on an ongoing basis, (c) it provides the beneficiaries a balanced financial flow, in which the income can meet daily basic needs and can be further invested, (d) it provides beneficiaries with the necessary skills to maintain or improve their quality of life through increased productivity. Integration of various stakeholder roles is also needed to achieve sustainability [25, 62].

## Concept of program performance

The program builds from a need to be relevant to the program's purpose, and the results will meet those needs. An active movement can achieve the program's objectives by designing a logic model. The logic model is a program implementation structure/scheme that requires a connection between the purposes and the program implementation process, which includes inputs (resources), activities, outputs, and actual outcomes in the short and long term [63]. An assessment of program performance needs to be done for evaluation, ex-ante (before implementation), and post-ex. This assessment involves internal institutions, experts, respondents [64], beneficiaries, and those who do not receive the program. The measurement uses a nominal, ordinal, interval, and ratio scale [63].

The ANP and MWI methods are used in this study to evaluate the performance of the Islamic Social Finance integration program. In this study, performance analysis of Islamic Social Finance integration is critical, with the following considerations: First, no research has been conducted to assess Islamic Social Finance integration programs with a focus on sustainability; Second, performance analysis is critical in determining the extent of the impact of a given program as well as decision-making material in program improvement.

## Integrated Islamic Social Finance

Integration is merging several parts into a unified whole [64]. Integration is also a collaboration between parties to achieve a common goal. Several previous researchers have analyzed the integrated Islamic Social Finance instruments. Jouti [25] developed a theoretical scheme by combining Islamic Social Finance instruments, such as zakat, waqf, sharia microfinance, and sukuk. Sulistyowati et al. [65] developed an integrated Islamic Social Finance model to address disaster-related problems. Haneef et al. [47] utilized the SEM analysis on integrated ISF (waqf and microfinance) and examined the relationship between takaful, waqf, and human resource development. Hassanain [66] developed three integration models combining zakat, waqf, and sharia microfinance. Previous research has remained focused on conceptual models and failed to demonstrate integration's efficacy in improving well-being.

Previous research has also proven that ISF integration positively impacts ISF management. Amuda [67] investigated the role and effectiveness of cash waqf, zakat, alms, and public funding in alleviating poverty in Muslim communities in Nigeria. The study showed that if managed efficiently, the integration of cash waqf, zakat, alms, and public funds can significantly contribute to community empowerment. Haneef et al. [47] developed waqf-based Islamic microfinance (IsMF) system to address poverty in Bangladesh. The study found that the integrated waqf and Islamic microfinance model could help reduce poverty in the community. Furthermore, according to Jouti [25], integrating several ISF instruments is one of the prominent factors in developing the ISF ecosystem. Widiastuti et al. [22] analyzed Islamic social finance integration solutions and strategies, showing that data integration is the most priority solution.

The Islamic Social Finance ecosystem can foster integration among Islamic Social Funds. Several Islamic social funds integration programs have been implemented in Indonesia. For example, the Ahmad Wardi Hospital program was built on waqf land and funded by zakat and infaq funds. As a result, mustahiq can freely use the health program. Furthermore, several Zakat Institutions in Indonesia run agricultural programs in which the land is provided by waqf lands, such as the programs run by the Zakat Institutions Al-Azhar and the Infaq Management Board. During the farming season, operational needs are met using infaq funds through the qardhul hasan scheme. This integration program can significantly accelerate community welfare and maximize the potential of Islamic social funds.

### Previous studies on Islamic Social Finance index

Several previous studies have built Islamic Social Finance Indexes, as shown in Table 2.

Previously, research has focused on measuring the institution's performance by focusing on a single aspect of measurements, such as service quality, accountability, or management efficiency. Furthermore, most previous studies used qualitative techniques, such as the Analytical Network Process (ANP) technique, without quantifying the indicators. As a result, this study built on previous research by developing a novel measurement that focuses on sustainability factors to assess the sustainability of Islamic Social Finance programs. Furthermore, this study uses the developed measure to determine the success of three empowerment programs implemented by various Islamic Social Finance Institutions.

## Material and method

### Method and technique analysis

The research method is separated into two phases. The first method stage is a qualitative approach using the Analytical Network Process (ANP) to formulate the index components of Islamic Social Finance programs' success and sustainability. The Analytical Network Process is a development of the Analytical Hierarchy Process, built by Saaty [76]. The researcher chose the ANP method with several considerations [38, 77–81]: (a) ANP is a method of multi-criteria decision making that can overcome complex network structures, (b) excellent in evaluating all

**Table 2. Prior Studies of Islamic Social Finance index.**

| No | Author | Purpose | Method |
|---|---|---|---|
| 1 | Wahab et al. [68] | This study develops a service quality index for Zakat Institutions | SEM |
| 2 | Bhanot and Bapat [69] | This study creates a sustainability index for microfinance institutions | Technique of Order Preference by Similarity to Ideal Solution (TOPSIS) |
| 3 | Kusuma and Ryandono [70] | This study expands the zakat index as a monitoring tool for zakat payments in Muslim majority countries | Content analysis |
| 4 | Abdullah et al. [29] | This study proposes Zakat Effectiveness Index (ZEIN) | Qualitative research |
| 5 | Ismail and Jabeen [71] | This study presents an integrated disclosure waqf index to evaluate established waqf property | Interpretive phenomenological analysis |
| 6 | Pyeman [72] | This study evaluates the efficiency of waqf using by Waqf Management Efficiency Index (WMEI) | Data Envelopment Analysis |
| 7 | Center of Strategic Studies BAZNAS [73] | This study offers the Zakat Utilization Index (IPZ) with five primary dimensions: social, cultural, economic, da'wah, and environment | Multistage weighted Index |
| 8 | Center of Strategic Studies BAZNAS [74] | This study proposes the Zakat Village Index as an assessment of the feasibility of a village to be given an empowerment program | Multistage weighted Index |
| 9 | Wahyudin et al. [75] | This study presents the National Zakat Index from micro and macro dimensions | Multistage weighted Index |

Source: Compiled by Authors

relationships between clusters and elements, and (c) ability to determine priorities of all proposed indicators.

There are three stages in the ANP method. The first is the decomposition stage, which aims to build and validate the framework of the ANP model by determining the objectives, factors, aspects, and indicators that influence the success and sustainability of Islamic Social Finance programs (see Fig 1). The results of Focus Group Discussion (FGD) and in-depth Interviews with Islamic Social Finance experts from representatives of academics, associations, regulators, and practitioners develop the ANP framework. The purpose is to obtain information on program management from various perspectives [82]. The study conducted three FGDs and two in-depth interviews. From the first and second FGDs, the researchers obtained an overview of successful and sustainable program management. The results of FGD 1 and 2 built the framework of the ANP model. The study also acquired previous research from reputable international journals Scopus Q1-Q3 and Top Tier to construct the model framework. Then, the third FGD and in-depth interviews validated the built framework.

The second ANP stage is a pairwise comparison to create the designed questionnaire from the ANP model. The questionnaire includes information on factors, stakeholders, aspects, and indicators that affect the success and sustainability of the program. The questionnaire was compiled based on the general ANP format using a scale of 1–9 for each compared point. A scale of 1 indicates very unimportant meaning; on the contrary, a scale of 9 means very important/relevant/influential. The questionnaires were distributed to experts to assess each ANP network.

The third stage is the analysis. It includes the priority weight of each index. The priority weight is determined following agreement results from experts. The calculation of priority weight uses the help of Microsoft Excel and Super Decision. Based on Saaty [79], the measure of each index is derived from the calculation of the Geometric Mean and Rater Agreement.

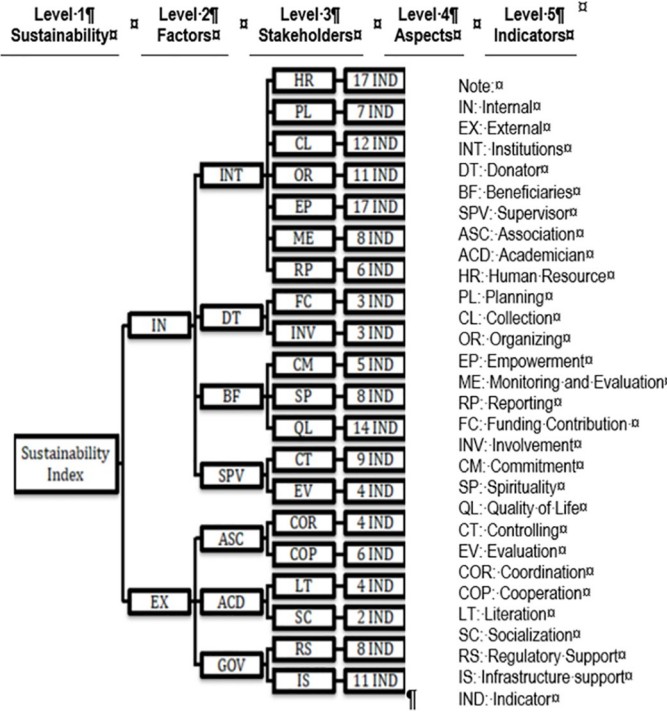

**Fig 1. Research framework of sustainability index of Islamic Social Finance program.** Source: Authors' Own.

Geometric Mean is a calculation of opinion in a cluster from expert assessments. On the other hand, the Rater Agreement (W) is the level of agreement of the experts in assessing the weight where the value of W = 1 indicates a perfect agreement. The formula for calculating the Geometric Mean and Rater Agreement is as follows:

$$GM = (R1 * R2 * R3 * R4 * \dots * Rn)1/n \tag{1}$$

Where R is respondent and n is number of respondents. Meanwhile,

$$W = \frac{S}{Max\ S} \tag{2}$$

$$S = \left(T1 - \left(\frac{T1 + \dots .Tp}{p}\right)^2 + \dots + Tp - \left(\frac{T1 + \dots + Tp}{p}\right)^2 \right. \tag{2.1}$$

$$Max\ S = \left(n - \left(\frac{T1 + \dots + Tp}{p}\right)^2\right)^2 + \dots + \left(pn - \left(\frac{T1 + \dots + Tp}{p}\right)^2\right)^2 \tag{2.2}$$

T is the transpose matrix from the results of data processing, p is the number of nodes, n is the number of respondents and S is the number of squared deviations from the calculation results of T and the average value of the absolute priority.

In detail, Fig 2 shows the stages of the ANP and MWI methods.

After all the phases in the ANP method have been carried out, the next step is to estimate the index value using the Multi Stage Weighted Index method (phase 2). Like the ANP method, the MWI method has three stages. The first stage is to develop criteria for each built indicator in the ANP model. The standards were compiled based on the results of the FGD, in-depth interviews, and literature reviews. The assessment of criteria is described using a Likert scale of 1–5, where 1 indicates the least ideal condition, while 5 shows the most optimal condition. The second stage is determining the weight for each factor and aspect to formulate the

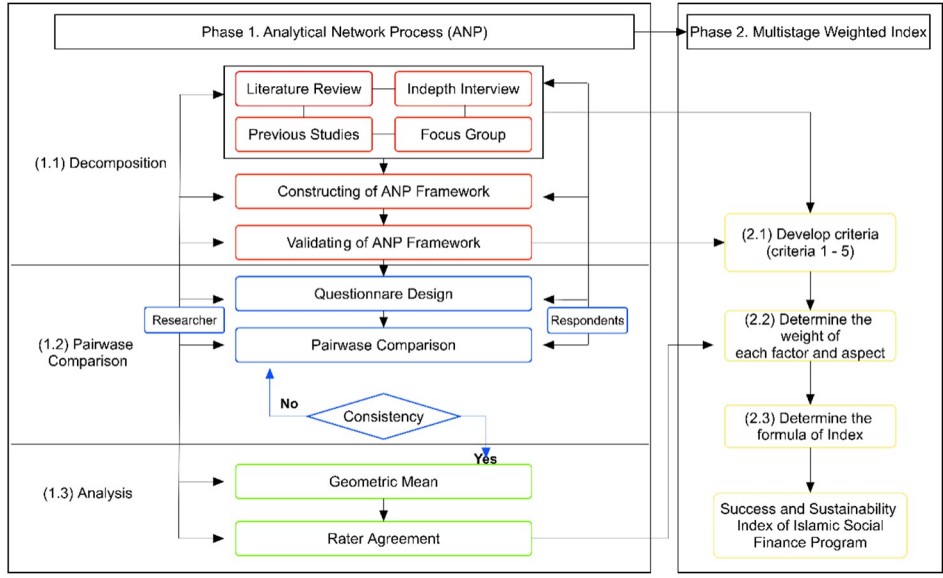

**Fig 2. Research method stages.**

index calculation. In this phase, there is a relationship between ANP and MWI where the weights' measurement uses the ANP data processing results. The third stage is to estimate the index value with the following formula:

$$SSI = \left( \left( \sum (MWI\ IND * NVA) \right) * NVS \right) * NVF)/5)$$ (3)

Where MWI IND is the average score of MWI score from each stakeholder, NVA is Normalized Value of Aspect, NVS is Normalized Value of Stakeholder, and NVF is Normalized Value of Factor.

This study built an index of the success and sustainability of Islamic Social Finance programs using the score of 0 to 100. In addition, the index is able to formulate the possibility of sustainability, as shown in Table 3.

## Data and samples

Primary data sources are focus group discussions, in-depth interviews, and field studies. The results of focus group discussions and in-depth interviews formulated the ANP model and criteria assessment for the MWI method. This study involved a total of 15 respondents consisting of four representatives of academics, four representatives of practitioners, three representatives of associations, and four representatives of regulators. In the ANP method, the number of respondents is not the main criterion but the expertise and experience [84]. The respondents filled out the questionnaire through several procedures. First, the researcher team sent an application letter to fill out a questionnaire. After receiving a response, the research team scheduled the time with the respondents. Second, the respondents were given several choices in filling out the questionnaire: (a) assistance from the research team through zoom meetings, (b) offline, and (c) independently. The research team structured the questionnaire in easy-to-understand sentences to avoid misinformation in the distributed questionnaires and provided a glossary for specific words/sentences. Full access to the questionnaire can be accessed by contacting the corresponding author. The data that can be accessed are (1) research questionnaires using the ANP method, and (2) answers from respondents to the questionnaire.

The selection of respondents used the purposive sampling technique. The respondents must had more than two publications in reputable international journals related to Islamic

**Table 3. Success and sustainability index of Islamic Social Finance program.**

| Success Index of Program | Sustainability Index of Program |
|---|---|
| 0–25, economic rescue | 0–8.33, low economic rescue |
|  | 8.34–16.67, medium economic rescue |
|  | 16.68–25, high economic rescue |
| 26–50, economic recovery | 26–34.33, low economic recovery |
|  | 34.34–42.67, medium economic recovery |
|  | 42.68–50, high economic recovery |
| 51–75, economic reinforcement | 51–59.33, low economic reinforcement |
|  | 59.34–67.67, medium economic reinforcement |
|  | 67.68–75, high economic reinforcement |
| 76–100, economic resilience / welfare | 76–84.33, low sustainability |
|  | 84.34–92.67, medium sustainability |
|  | 92.68–100, high sustainability |

Source: Authors' compilation

Social Finance for the academic group. In the group of practitioners, respondents were in managerial positions as chairpersons or program managers. The respondents were leaders in the Islamic economic community association and the Islamic Social Finance management forum for association groups. Next, the respondents were from the Indonesian Waqf Board, the National Amil Zakat Agency, the Ministry of Religion, and KNEKS and held managerial positions for the regulator group.

On the other hand, field studies present the empirical results of the constructed index of success and sustainability. The study analyzed three Islamic Social Finance institutions with the following criteria: (a) official institutions that are granted a license to operate by the government, (b) have the status of a national or provincial level institution, and (c) conduct empowerment programs. Furthermore, the empowerment program's criteria must include the following terms. First, it is an Islamic Social Finance integration program. The funds disbursed are not only from one type of source. Second, the program has run for at least one year. Third, the program implements benefits to mustahiq, who deserve to be empowered. Similar to the procedure for collecting questionnaire data with experts, the researcher also confirmed the willingness of the Islamic Social Finance Institution to become research respondents. Data entry and interviews with representatives from ISF institutions were managed offline (2 respondents) and online (1 respondent). Before taking the questionnaire, experts had to fill out a consent form. Each expert completed the questionnaire consciously and in a normal mental state. Further, the experts were not subjected to any pressure from any party.

This research follows the rules and ethics of research conduct and writing by the Institute for Research and Community Service (LPPM) of Universitas Airlangga. This research also refers to Airlangga Chancellor's Regulation Number 34 of 2019 regarding the rules of behavior in article 16 (b); researchers must be honest, objective, and pay attention to all aspects of the research process and are not allowed to manipulate data and research results. Article 16 (f) states that researchers must respect and appreciate the object of research, whether the thing is a human or an animal, whether living or dead. Further, this research is supervised by the Center of Research and Publication (3P) in the Faculty of Economics and Business Universitas Airlangga, which functions as an ethics committee according to the Dean Decree Number 88/UN3.1.4/2020. This research is also supervised by the Institute for Research and Community Service (LPPM) Universitas Airlangga as the ethics committee at the university level and as a funder.

## Concept of 4 ER (economic rescue, recovery, reinforcement, resilience)

The 4 ER concept is a concept that shows the extent to which the Islamic Social Finance program can prosper the beneficiaries. The researchers of this study created the concept based on literature reviews on prior studies and previously conducted focus group discussions. The 4 ER concept was developed based on the maqashid sharia. According to Al-Ghazali, maqashid sharia (objectives of Islamic Law) aims to achieve maslahah (means welfare/advantage/merit) by maintaining five aspects: Faith, Soul, Mind, Lineage, and Prosperity. This study used the research of Kusuma and Ryandono [70] as the basis for constructing the 4 ER. Kusuma and Ryandono [70] determined the economic category of society, as shown in Fig 3. In Fig 3, the economy of society is divided into 3, namely, (1) underprivileged and entitled to receive zakat ($0\text{-}Y_{KOMZ} = C_0$), (2) pre prosperous but not entitled to receive zakah nor must pay zakat ($Y_{KOMZ} = C_0$), and (3) prosperous and to pay zakat ($Y_{ON} = C_{ON}$). Thus, the researcher developed indicators in these five aspects and determined the level of concept 4 ER, as shown in Table 3.

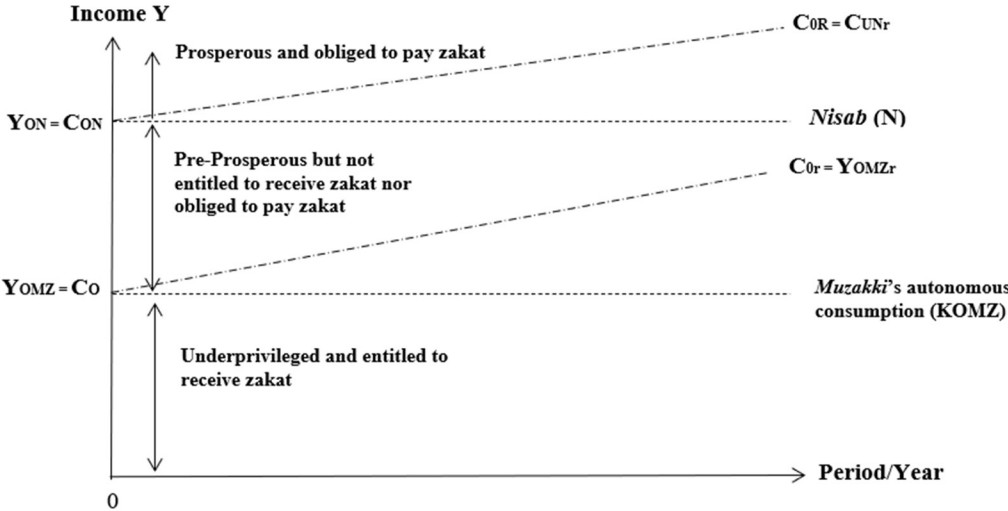

**Fig 3. Level of economic society.** Source: Kumara and Ryandono [70].

The first is Economic Rescue, the lowest level, where the Islamic Social Finance program cannot improve the beneficiaries' economy ($Y < S$, no job). This level means that the program has not been able to encourage recipients to be able to meet the needs in the five aspects of maqashid sharia. The value score of the Economic Rescue is 1–25%. Secondly, Economic Recovery was defined as the level where the program can provide economic change and encourage beneficiaries to meet their needs, but it is not a significant change. In Economic Recovery, program recipients have not been able to save their income ($Y < S$), but the most basic needs have been fulfilled, although it is unstable. According to Chang and Rose [83], economic recovery is an activity to recover economic conditions from a disaster. The score for Economic Recovery is 26–50%.

Next, Economic Reinforcement is the third level, showing that the Islamic Social Finance program can increase the beneficiaries' economy, and the income exceeds the needed ($Y > S$). At this stage, the program's recipients have been able to save, invest, and donate their income. The score for Economic Recovery is 51–75%. Lastly, Economic Resilience is the top level where the program has created prosperity for its beneficiaries. According to Shima et al. [84], economic resilience can be defined as transforming and improving an organization's performance. This stage is considered economic resilience if the program's recipients can save income, grow business assets, and be able to pay zakat. The score for Economic Reinforcement is 76–100%. This study developed the concept of 4 ER by specifying a more detailed level: Low, middle, high, and sustainable. The program is sustainable if it has achieved Economic Resilience. At this level, the researcher divided the sustainability level into low, middle, and high.

## Result and discussion

### Index of success and sustainability of Islamic Social Finance program

This study built an index of the success and sustainability of Islamic Social Finance programs. The index was constructed by analyzing factors, stakeholders, aspects, and indicators compiled through focus group discussions, in-depth interviews, and literature reviews. Table 4 shows the weights of each index component processed using the ANP method.

**Table 4. Data processing results of success and sustainability index of Islamic Social Finance program.**

| Factor | Normalized Value | Stakeholder | Normalized Value | Aspect | Normalized by Aspect | Normalized of Aspect per Factor | Rank |
|---|---|---|---|---|---|---|---|
| Internal | 0.5 | Islamic Social Finance Institutions | 0.38 | Human Resource | 0.19 | 0.03 | 8 |
| | | | | Planning | 0.16 | 0.02 | 9 |
| | | | | Collection | 0.15 | 0.02 | 9 |
| | | | | Organizing | 0.14 | 0.02 | 9 |
| | | | | Empowerment | 0.14 | 0.02 | 9 |
| | | | | Monitoring and Evaluation | 0.10 | 0.02 | 9 |
| | | | | Reporting | 0.12 | 0.02 | 9 |
| | | Donator | 0.24 | Funding Contribution | 0.71 | 0.22 | 1 |
| | | | | Involvement | 0.29 | 0.09 | 2 |
| | | Beneficiaries | 0.24 | Commitment and Involvement | 0.40 | 0.06 | 5 |
| | | | | Spirituality | 0.23 | 0.03 | 8 |
| | | | | Quality of Life | 0.36 | 0.05 | 6 |
| | | Supervisor | 0.14 | Controlling | 0.56 | 0.08 | 3 |
| | | | | Evaluation | 0.44 | 0.06 | 5 |
| External | 0.5 | Academician | 0.29 | Literation | 0.55 | 0.07 | 4 |
| | | | | Socialization | 0.45 | 0.06 | 5 |
| | | Association | 0.28 | Cooperation | 0.48 | 0.06 | 5 |
| | | | | Coordination | 0.52 | 0.07 | 4 |
| | | Government | 0.43 | Policy and Regulation | 0.55 | 0.05 | 6 |
| | | | | Infrastructure support | 0.45 | 0.04 | 7 |
| **TOTAL** | | | | | | **1** | |

Sources: Data Primary Result (2022)

## Factor and involved stakeholder

Internal and external factors influencing the ISF program's success show the same important value (0.5). Four stakeholders play a role in internal factors, including institutions (0.38), donors (0.24), beneficiaries (0.24), and supervisors (0.14) (see Table 4). Islamic Social Finance institutions (0.38) are the most crucial stakeholder in the program's success. Islamic Social Finance institutions must be professional in creating and developing programs supported by competent human resources [5, 85], proper program planning, ability to collect, manage and empower, conduct systematic monitoring and evaluation, and perform transparency of program [86–88]. On the other hand, the critical roles of donors, beneficiaries, and supervisors help achieve the program's success and sustainability. Donors have an important role in channeling funds to institutions [26, 60, 89, 90]. Beneficiaries are essential stakeholders because Islamic Social Finance programs aim to improve the beneficiaries' quality of life [6, 14]. Meanwhile, the supervisor ensures that the program does not violate the applicable laws and regulations [91].

Meanwhile, the external stakeholders include academicians (0.29), associations (0.28), and the government (0.43). The government is the most influential stakeholder in the success and sustainability of the program. The government is the party that permits the establishment of institutions and implementation of Islamic Social Finance programs [92]. Moreover, the government is the decision-maker regarding issuing regulations and policies that support the success and sustainability of the program [5, 93–95]. On the other hand, academics and

associations also play an essential role. Academics are responsible for developing competent human resources and increasing literacy in Islamic Social Finance. Associations play a role in collaborating and coordinating with all parties to create an ecosystem that supports the success and sustainability of the program.

## Aspect of sustainability program

Table 4 shows the results of the data processing for each aspect of each stakeholder that supports the success and sustainability of the Islamic Social Finance program. Overall, funding contribution from donors (0.22) is the priority in supporting aspects of the success and sustainability of the program. The result follows the concept of sustainability in microenterprise proposed by Pischke [59], which is balanced incoming and outgoing funds. In Islamic Social Finance, incoming funds come from donors/muzakki who pay zakat/infaq/alms/waqf to the institution, which will then be managed and distributed to beneficiaries. The result is reinforced by a prior study's finding [58] that the limited funds distributed to recipients caused the program's failure. The more funds raised, the more allocated amount given for the welfare of the beneficiaries. The effort is to achieve the welfarist paradigm in sustainability, proven by the increasing number of beneficiaries [26]. Islamic Social Finance institutions also play an essential role in encouraging the involvement of donors to pay the funds. Thus, Islamic Social Finance institutions need to design various strategies that increase the literacy and interest of donors to donate their funds through the institution. If donors prefer to pay outside the institution, it may result in the unmanageability of professional, productive, and sustainable funds.

The second priority is the involvement of donors (0.9) in providing input/advice to Islamic social finance institutions. This involvement includes the ability to capture donors' aspirations, trust, and loyalty to pay Islamic social funds (zakat, waqf, infaq, alms, and so on) to the institution [5, 96, 97]. Controlling (0.08) is the third priority aspect that supports the success and sustainability of the program, as supported by Ghosh et al. [98]. The responsibility for distributing Islamic social funds includes legal and sharia duties. Thus, control from the supervisor is necessary to ensure that the program does not violate the legal and sharia rules. The controlling aspect can also increase program transparency [99].

## Empirical study

This study attempted to measure the index of success and sustainability of Islamic Social Finance programs using three programs from different institutions represented by initials as programs A, B, and C. Table 5 shows the descriptions of the empowerment programs in this study.

Table 5 describes the types and objectives of the empowerment programs analyzed in this study. The programs are integrated, in which the funds distributed to recipients are not only from one Islamic social fund source but also more. Programs A and C have the same empowerment focus area: rice farming. At the same time, program B focuses on mushroom farming. After the analysis, Table 6 was generated to show the success of the programs and their sustainability index scores.

Based on Table 6, the success rate of program A in transforming mustahiq into muzakki is at the level of Economic Resilience with a score of 81% (see Table 3). Program A has the potential to create sustainability for its beneficiaries even though it is still at a low level of sustainability (see Table 3). On the other hand, Program B has an index score of 73%, which indicates the program's success is at the level of Economic Reinforcement. In this study, Economic Reinforcement means that the program has improved the beneficiaries' economy but has not yet reached the level of prosperity. The achieved level implies income and expenses of the mustahiq are still balanced. Next, program B has not yet achieved sustainability but has gained high

**Table 5. Islamic Social Finance programs.**

| | A | B | C |
|---|---|---|---|
| **Types and Objectives** | This program empowers farmer groups through business capital, training, and mentoring. Farmers were given training on Healthy Crop Management (HCM), where the program provider gives rice plants natural fertilizers and reduces chemical fertilizers. | This program is an empowerment program for the poor (*dhuafa*). A group of *dhuafa* was given business capital as a Mushroom House to farm. | This program focuses on agriculture. The program's beneficiaries receive assistance in business capital and subsidy. |
| | In addition to producing high-quality rice, HCM aims to increase the number of harvests in one year. | | During the program, the beneficiaries also receive training and assistance related to green farming. |
| | Generally, farmers will only have two harvests in one year. Meanwhile, with HCM, farmers can harvest three times in one year. Thus, farmers' income can increase. | | |
| **Integrated Islamic Social Funds** | Zakat, Infaq, and Waqf | Infaq and Sadaqah | Waqf and Infaq |
| **Beneficiary** | Farmers Group consists of *Dhuafa*** | *Dhuafa*** | Farmers Group consists of *Dhuafa*** |
| | **Dhuafa means someone who lives in poverty, disability, and powerlessness. | | |
| **Program Year** | 2020 | 2021 | 2019 |

Source: Authors' Compilation (2022)

Economic Reinforcement (see Table 3). Program C scored 80% and reached the level of Economic Resilience with low sustainability.

Fig 4 shows the index scores for each stakeholder in three different programs. In Islamic Social Finance institutions, aspects of planning and human resources have the highest scores in achieving success and sustainability in programs A and B. The planning aspect relates to how Islamic Social Finance institutions plan programs carefully by developing short-term and long-term strategies and formulating risk mitigation [26, 60]. The institutions are also responsible for planning an exit strategy to create financial self-sufficiency for beneficiaries and eliminate dependence on constant assistance. Proper program planning must be supported by the availability of competent human resources [5, 91, 100, 101]. Human resources have an essential role in creating innovative and strategic programs. There are various problems in the implementation of programs. One of them is differences in the beneficiaries' environment [58], the level of motivation to self-develop [5], and other problems that require human resources to build strategies so that the program runs optimally and creates maximum sustainable benefits. Further, the management and collection aspects have the highest scores in achieving success and sustainability in program C. In implementing the program, there must be good governance [99, 102–105]. Like the collection aspect, Islamic Social Finance needs an innovative collection strategy, an accurate database, and an excellent optimal service [68, 106, 107] to support donors' loyalty.

From the donor aspect, program C scores higher than the other programs. Donors have an active role in the success and sustainability of program C to transform beneficiaries' welfare. The involvement of donors, primarily in distributing Islamic social funds to institutions, is crucial in achieving the success and sustainability of the program. This finding follows prior studies [26, 59, 60], which mentioned the institutionalist and welfarist paradigms in sustainability. In the context of Islamic Social Finance, the institutionalist paradigm concerns how Islamic social funds continue to be collected, managed, and distributed to recipients in empowerment programs. The welfarist paradigm happens when there is an expansion and addition of communities that can improve their welfare through empowerment. Thus, the

**Table 6. Empirical investigation of Islamic Social Finance program.**

| | | | | | | MWI Score of Each Aspects | | | Index Score of Each Aspects | | | Index Score of Each Stakeholders | | | Indexx Score of Each Factors | | |
|---|---|---|---|---|---|---|---|---|---|---|---|---|---|---|---|---|---|
| Factor | NV | Stakeholder | NV | Aspect | NV | A | B | C | A | B | C | A | B | C | A | B | C |
| Internal | 0.5 | Islamic Social Finance Institutions | 0.38 | Human Resource | 0.19 | 3.76 | 3.76 | 3.06 | 0.73 | 0.73 | 0.59 | 1.69 | 1.55 | 1.52 | 2.07 | 1.94 | 2.15 |
| | | | | Planning | 0.16 | 4.86 | 4.14 | 3.71 | 0.75 | 0.64 | 0.58 | | | | | | |
| | | | | Collection | 0.15 | 4.33 | 4.08 | 4.33 | 0.65 | 0.61 | 0.65 | | | | | | |
| | | | | Organizing | 0.14 | 4.73 | 4.45 | 4.91 | 0.66 | 0.62 | 0.69 | | | | | | |
| | | | | Empowerment | 0.14 | 4.35 | 4.00 | 4.00 | 0.61 | 0.56 | 0.56 | | | | | | |
| | | | | Monitoring and Evaluation | 0.10 | 4.88 | 4.38 | 4.63 | 0.50 | 0.45 | 0.47 | | | | | | |
| | | | | Reporting | 0.12 | 4.67 | 4.00 | 3.83 | 0.56 | 0.48 | 0.46 | | | | | | |
| | | Donors | 0.24 | Contribution | 0.71 | 3.67 | 3.67 | 5.00 | 2.62 | 2.62 | 3.57 | 0.83 | 0.90 | 1.19 | | | |
| | | | | Involvement | 0.29 | 3.00 | 4.00 | 5.00 | 0.86 | 1.15 | 1.43 | | | | | | |
| | | Beneficiaries | 0.24 | Commit and Involvement | 0.40 | 4.40 | 3.20 | 3.80 | 1.78 | 1.30 | 1.54 | 0.96 | 0.80 | 0.92 | | | |
| | | | | Spirituality | 0.23 | 4.25 | 3.88 | 4.50 | 1.00 | 0.91 | 1.05 | | | | | | |
| | | | | Quality of Life | 0.36 | 3.50 | 3.14 | 3.50 | 1.26 | 1.13 | 1.26 | | | | | | |
| | | Supervisor | 0.14 | Controlling | 0.56 | 5.00 | 4.78 | 4.89 | 2.82 | 2.70 | 2.76 | 0.66 | 0.62 | 0.68 | | | |
| | | | | Evaluation | 0.44 | 4.50 | 4.25 | 5.00 | 1.96 | 1.85 | 2.18 | | | | | | |
| Eksternal | 0.5 | Association | 0.28 | Cooperation | 0.48 | 4.00 | 3.50 | 3.33 | 1.94 | 1.70 | 1.62 | 1.27 | 1.02 | 1.04 | 1.99 | 1.72 | 1.86 |
| | | | | Coordination | 0.52 | 5.00 | 3.75 | 4.00 | 2.58 | 1.93 | 2.06 | | | | | | |
| | | Academician | 0.29 | Literation | 0.55 | 3.00 | 4.75 | 4.50 | 1.65 | 2.61 | 2.48 | 0.80 | 1.21 | 1.24 | | | |
| | | | | Socialization | 0.45 | 2.50 | 3.50 | 4.00 | 1.12 | 1.57 | 1.80 | | | | | | |
| | | Government | 0.43 | Policy and Regulation | 0.55 | 4.50 | 2.88 | 3.50 | 2.50 | 1.59 | 1.94 | 1.91 | 1.21 | 1.44 | | | |
| | | | | infrastructure support | 0.45 | 4.36 | 2.73 | 3.18 | 1.94 | 1.21 | 1.42 | | | | | | |
| SUM Index of Internal and External Factor | | | | | | | | | | | | | | | 4.06 | 3.66 | 4.01 |
| Final Score of Success and Sustainability of Islamic Social Finance Program Index | | | | | | | | | | | | | | | 0.81 | 0.73 | 0.80 |

Note: NVF: Normalized Value of Factor, NVS: Normalized Value of Stakeholder, NVA: Normalized Value of Aspect, A, B, C: Code of three different Islamic Social Finance Empowerment Programs

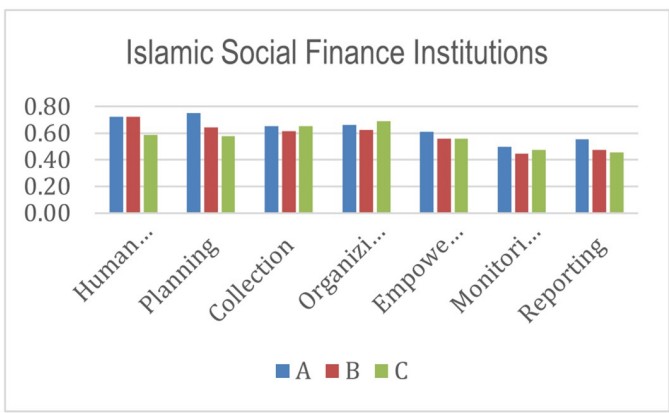

**Fig 4. Index score of Islamic Social Finance institutions.**

involvement and loyalty of donors to channel Islamic social funds into Islamic Social Finance institutions is crucial [96, 108, 109].

From the beneficiary's perspective, commitment and motivation to change is the highest aspect of all programs' success and sustainability. This result is relevant to the previous study [5, 110, 111]. There should be a commitment from the beneficiaries to improve their welfare condition so that empowerment programs can be productive [5, 110]. Scores for aspects of spirituality and quality of life are still low. The insignificant score explains why programs A, B, and C are still at Low Sustainability. However, beneficiaries of programs A, B, and C have a high commitment to change.

The controlling aspect value is higher than the evaluation aspect for the entire program by seeing the supervisor stakeholder. The result means the controlling element has been carried out but not thoroughly evaluated. The coordination aspect score is higher than the cooperation for all programs from the association stakeholder's perspective. Coordination includes how associations coordinate with all stakeholders and build an ecosystem that supports program implementation [25]. Program A has the highest score than programs B and C. In other words, program A gets more support from the association in achieving the success and sustainability of the program.

From the academician perspective, programs B and C have identical scores and are higher than program A. The result means that programs B and C have full support from academics in program implementation. Academic support, in this case, is in the form of involvement in providing competent human resources and socializing the importance of Islamic Social Finance, increasing public literacy.

Seen from the government stakeholder, the policy and regulation aspect is highly valued compared to the infrastructure support aspect in all programs. Program A has the highest score in policy and regulation compared to programs B and C. Based on the interviews with program A managers, the government collaborated with Islamic Social Institutions for program A implementation. The government supported by providing funds and inputs to optimize the execution. The government invited other stakeholders to discuss the direction of program A development. Policy and regulatory approval from the government are fundamental to achieving the success and sustainability of the program [22, 101, 112]. The high score may explain why program A is more successful and sustainable than programs B and C.

From the histograms shown in Figs 4–10, program A becomes the program with the highest index value (81%) among all programs. It excels in several aspects, such as planning, human

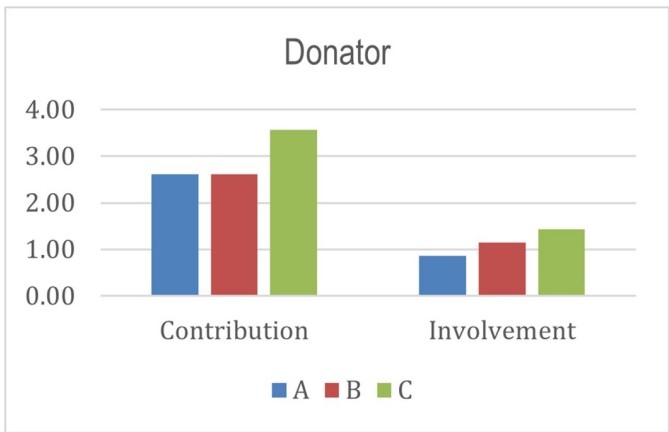

**Fig 5. Index score of donator.**

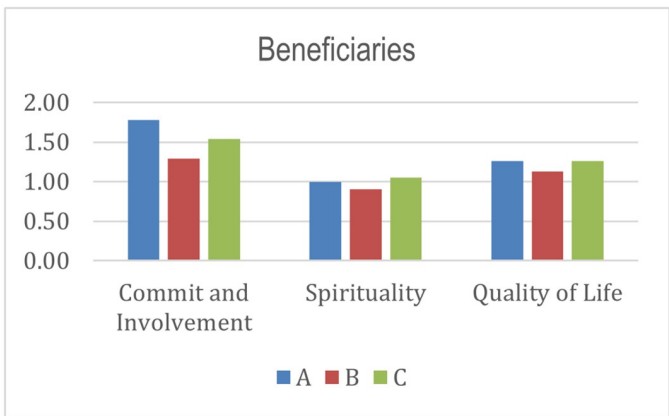

**Fig 6. Index score of beneficiaries.**

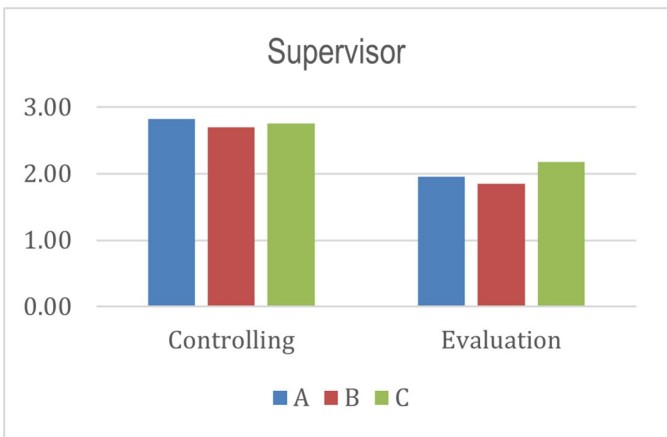

**Fig 7. Index score of supervisor.**

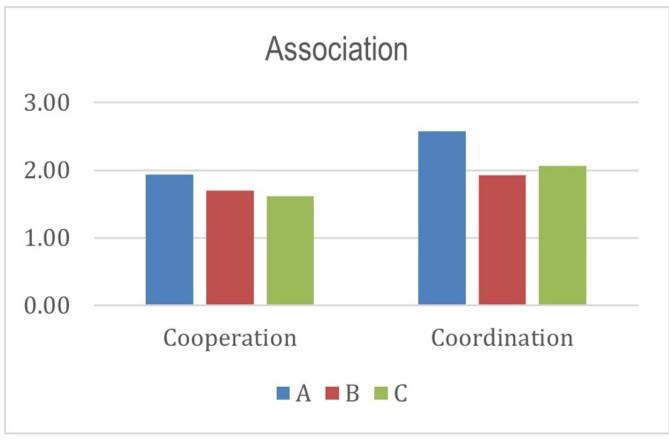

**Fig 8. Index score of association.**

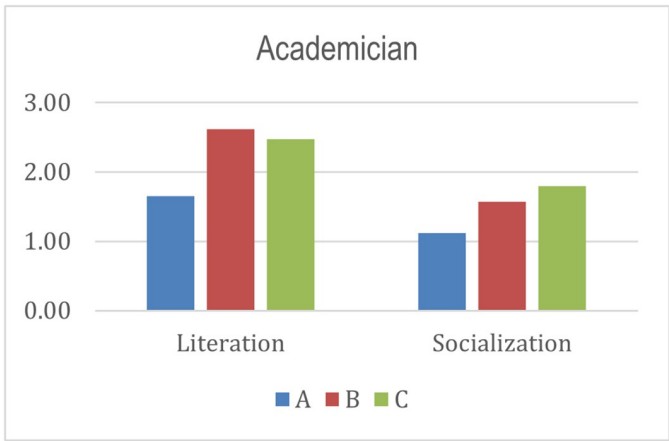

**Fig 9. Index score of academician.**

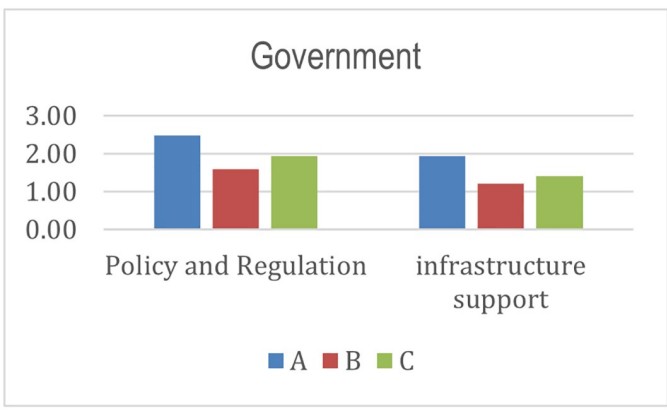

**Fig 10. Index score of government.**

resources, commitment and involvement of beneficiaries, beneficiaries' quality of life, controlling, coordination, cooperation, policy and regulation, and infrastructure support. The result means program A fulfills various important indicators in achieving the success and sustainability of the program. However, program A must improve coordination with academicians to raise public awareness about Islamic social funds and strengthen its governance. Further, Program A must increase donor loyalty to improve the amount of Islamic social funds distributed to beneficiaries.

Program B has the lowest score and is on the economic reinforcement level. Compared to programs A and C, Program B must optimize coordination and teamwork with all stakeholders, particularly beneficiaries and the government. In addition, program B must increase beneficiary commitment and responsibility and develop strategies to improve beneficiaries' quality of life. Furthermore, program B must carry out synergy and coordination with the government, focusing on the synergy of assisting the government in program implementation. Program C has achieved the same level of success and sustainability as Program A. In general, program C must optimize aspects involving internal ISF stakeholders.

This study analyzed three utilization programs so that they can be benchmarked against each other. The index helps Islamic Social Finance institutions to manage empowerment

programs by tracking the level of success and sustainability of the program. The purpose is to understand the extent of the empowerment programs' impact on welfare transformation on program recipients.

## Theoretical implications

The study adds to the literature by developing an index for measuring the performance of the ISF Institution by emphasizing the aspects of sustainability of the empowerment programs in transforming the welfare of mustahiq. The established index provides an exact measure of success and sustainability through the 4 ER concepts (Economic Rescue, Economic Recovery, Economic Reinforcement, Economic Resilience). Thus, the government and Islamic Social Finance institutions can use this index to measure the success and sustainability of a program. Further, this study adds to the literature regarding the integration theory of Islamic Social Finance in empowerment programs by implementing the index on three integrated programs in Indonesia.

## Practical implications

The study gives managerial contribution in several ways. First, the study emphasizes the importance of the involvement of donors in paying Islamic social funds to Institutions to be managed productively. Islamic Social Finance institutions must formulate strategies to increase donor engagement. One is by providing platforms that can create more manageable payments, such as e-commerce, crowdfunding, mobile banking, and so on, or build potential programs that have long-term impacts. Other way is using digital technology, including social media, to educate and socialize Islamic Social finance by displaying creative content as educational media. Collaboration with influential figures from each generation (baby boomers, millennials, gen Z, and so on) is critical because each generation has unique characteristics and approaches. As a result, the purpose of socialization can be effectively communicated, and the amount of fundraising can increase. More than 30 universities in Indonesia host lectures on Islamic Social Finance/Islamic Economics/Sharia Economics. As a result, academics play an essential role at the university level in socializing the importance of Islamic social fund donation. According to the findings of this study, the government is the most influential external stakeholder. As a regulator and leader of economic countries, the government issues policies and regulations advising the public to pay Islamic social funds through institutions. Furthermore, local governments, such as the Aceh district through Qanun Aceh No. 10 Years 2018, have obligated civil servants to pay zakat. However, this policy is still subject to each local government policies. With this, the government must reconsider the urgency of requiring the payment of Islamic social funds for Muslims through the ISF Institution nationally.

Second, an empirical investigation of the Islamic Social Finance programs shows that the programs are at the Economic Reinforcement and Economic Resilience levels with low sustainability. Islamic Social Finance institutions must develop short-, medium-, and long-term strategies to ensure mustahiq welfare. In the short term, Islamic Social Finance institutions must focus on strategies enabling mustahiq to meet their basic needs. The approach is purposed to alleviate distress and ensure the safety of the beneficiaries. In the medium term, Islamic Social Finance must develop a strategy to increase mustahiq's income and enable them to meet their basic needs independently through business management. The approach provides mustahiq with the necessary empowerment program to improve their entrepreneurial skills. Finally, in the long run, the Islamic Social Finance institution must devise a strategy to break mustahiq's reliance on the institution, allowing the mustahiq to be self-sufficient. Training and mentoring for mustahiq is one way to increase mustahiq's skills and reach prosperity.

The government can help by providing beneficiaries with training and mentoring. The support certainly relieves Islamic Social Finance institutions when some have difficulty giving training to their recipients. Furthermore, one of the causes of the program's low sustainability is the limited funds. The government must support Islamic Social Finance Institution in running the empowerment program. The government must assist Islamic Social Finance institutions in implementing the empowerment initiative. To create welfare for mustahiq, the institutions must optimize teamwork and coordination with various stakeholders, such as associations, private corporations, and others.

## Conclusion

This study developed a success and sustainability index for Islamic Social Finance Program using Analytical Network Process (ANP) and Multistage Weighted Index (MWI). The study analyzed various aspects and indicators to achieve program success and sustainability. The results show that the funding contribution of donors is a priority aspect. It indicates that more funds need to be paid by donors through institutions. The funds are then managed professionally in the form of empowerment programs. The programs must achieve success and sustainability to have a broader impact on the recipients. The realization of collections to institutions is still low, so the government must issue mandatory policies related to the payment of Islamic social funds in institutions. Therefore, Islamic Social Finance Institutions must manage the program professionally to transform welfare for the recipients with limited funds. This can be done by transferring knowledge and skills to beneficiaries through training and mentoring. In addition, this study highlights donors' involvement and supervisors' role in program monitoring to achieve program success and sustainability. This study conducted an empirical analysis of three empowerment programs in three different institutions. The finding shows that programs A (0.81) and C (0.80) have encouraged beneficiaries to improve their quality of life even though they are not economically stable. Meanwhile, program B (0.73) is at the level of economic reinforcement, which means the program has not yet achieved sustainability.

The study has limitations. First, the index built in this study is based on specific conditions in Indonesia and cannot be generalized to countries with other different characteristics of Islamic Social Finance programs. Thus, it is necessary to develop cross-country research to apply the index to different countries. Second, the empirical investigation only includes integrated programs. Hence, further analysis can conduct empirical studies on integration and non-integration programs to determine the urgency of Islamic Social Finance integration.

## Supporting information

**S1 Data.**
(XLSX)

**S1 File.**
(DOCX)

## Acknowledgments

The authors would like to express their gratitude to the Center Research and Publication (3P) of Faculty Economic and Business Universitas Airlangga and the Institute for Research and Community Service (LPPM) Universitas Airlangga as the ethics committee. The authors would like to thank representatives of academics, practitioners, regulators, and associations who have contributed to preparing the components of the program's success and sustainability index. The authors would also like to thank Islamic Social Finance Institutions for the

empirical measurement of the success and sustainability of their program. The authors would also like to extend gratitude to the Institute of Technology Management and Entrepreneurship, Universiti Teknikal Malaysia Melaka (UTeM), Melaka, Malaysia as an international research partner. The authors would also like to thank Mrs. Nikmatul Atiya, Mrs. Miratun Nisa, Mrs. Adelia Tsamara, Mrs. Siti Sahiroh, Mr. Dzikri Nurrohman, and Mr. Muhammad Hasdyani Putra as research team members for their support in the completion of this research.

## Author Contributions

**Conceptualization:** Tika Widiastuti.

**Formal analysis:** Arie Prasetyo, Imron Mawardi.

**Funding acquisition:** Tika Widiastuti.

**Investigation:** Arie Prasetyo, Imron Mawardi.

**Methodology:** Arie Prasetyo, Imron Mawardi, Rida Rosida, Muhammad Ubaidillah Al Mustofa.

**Project administration:** Anidah Robani.

**Resources:** Muhammad Ubaidillah Al Mustofa.

**Software:** Muhammad Ubaidillah Al Mustofa.

**Supervision:** Tika Widiastuti.

**Writing – original draft:** Imron Mawardi, Rida Rosida, Muhammad Ubaidillah Al Mustofa.

**Writing – review & editing:** Anidah Robani.

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
