## [Decision Letter · Decision Letter 0]

28 Aug 2022

PONE-D-22-20909Toward Developing a Sustainability Index for the Islamic Social Finance Program: An Empirical InvestigationPLOS ONE

Dear Dr Robani,

Thank you for submitting your manuscript to PLOS ONE. After careful consideration, we feel that it has merit but does not fully meet PLOS ONE’s publication criteria as it currently stands. Therefore, we invite you to submit a revised version of the manuscript that addresses the points raised during the review process.

We look forward to receiving your revised manuscript.

Kind regards,

Abdul Aziz Khan Niazi, Ph. D.

Academic Editor

PLOS ONE

Journal Requirements:

3. PLOS ONE does not copy edit accepted manuscripts (https://journals.plos.org/plosone/s/criteria-for-publication#loc-5). To that effect, please ensure that your submission is free of typos and grammatical errors.

5. Please ensure that you refer to Figure 2 in your text as, if accepted, production will need this reference to link the reader to the figure.

Additional Editor Comments:

Minor revision is required.

Reviewers' comments:

Reviewer's Responses to Questions

**Comments to the Author**

1. Is the manuscript technically sound, and do the data support the conclusions?

Reviewer #1: Yes

Reviewer #2: Partly

2. Has the statistical analysis been performed appropriately and rigorously? 

Reviewer #1: N/A

Reviewer #2: Yes

3. Have the authors made all data underlying the findings in their manuscript fully available?

Reviewer #1: Yes

Reviewer #2: Yes

4. Is the manuscript presented in an intelligible fashion and written in standard English?

Reviewer #1: Yes

Reviewer #2: No

5. Review Comments to the Author

Reviewer #1: This articles is very interesting and beautifully written, however few grammatical, punctuations and syntax issues were seen. This paper can be accepted. I recommend to add the time series of different policies and their impact on social financing program. In addition, please elaborate more on exiting system loops and holes and their long-term consequences on financing.

Reviewer #2: 1. Please rephrase the objectives of the study. I believe it will help readers to clearly understand the objective of the study.

2. A summary paragraph of literature review will help readers to understand the overall depiction of literature.

3. Please add more details in the discussion section.

4. In the conclusion section, please add limitations of the study and future recommendations. Please also provide clear policy implications. I suggest the author to write implications one by one so readers can easily read it.

Overall, the author should review the paper, remove all grammatical and punctuations related errors.

6. PLOS authors have the option to publish the peer review history of their article (what does this mean?). If published, this will include your full peer review and any attached files.

Reviewer #1: No

Reviewer #2: No

---

## [Author Response · Author response to Decision Letter 0]

21 Sep 2022

Sept 21, 2022

Today just uploaded the requested file Manuscript_No Track Changes.

Reviewer #1: This articles is very interesting and beautifully written, however few grammatical, punctuations and syntax issues were seen. This paper can be accepted. I recommend to add the time series of different policies and their impact on social financing program. In addition, please elaborate more on exiting system loops and holes and their long-term consequences on financing.

WE THANK THE REVIEWER FOR THIS CONSTRUCTIVE COMMENT. WE HAVE IMPROVED THE ARTICLE AS SUGGESTED. KINDLY REFER TO PAGE 19

Reviewer #2: 1. Please rephrase the objectives of the study. I believe it will help readers to clearly understand the objective of the study.

2. A summary paragraph of literature review will help readers to understand the overall depiction of literature.

3. Please add more details in the discussion section.

4. In the conclusion section, please add limitations of the study and future recommendations. Please also provide clear policy implications. I suggest the author to write implications one by one so readers can easily read it.

Overall, the author should review the paper, remove all grammatical and punctuations related errors.

THANK YOU FOR THE COMMENT. WE HAD GIVEN OUR BEST TO REVISE ACCORDINGLY AND HAD SENT THIS ARTICLE FOR PROFESSIONAL PROOF READING TO MINIMIZE THE TYPO AND GRAMMATICAL ERRORS. KINDLY REFER TO PAGES 1-2, 3-7, 18 AND 20 FOR DETAILS

---

## [Editor Report · Decision Letter 1]

17 Oct 2022

Toward Developing a Sustainability Index for the Islamic Social Finance Program: An Empirical Investigation

PONE-D-22-20909R1

Dear Dr. Anidah Robani,

We’re pleased to inform you that your manuscript has been judged scientifically suitable for publication and will be formally accepted for publication once it meets all outstanding technical requirements.

Kind regards,

Abdul Aziz Khan Niazi, Ph. D.

Academic Editor

PLOS ONE
---

## [Editor Report · Acceptance letter]

10 Nov 2022

PONE-D-22-20909R1 

Toward Developing a Sustainability Index for the Islamic Social Finance Program: An Empirical Investigation 

Dear Dr. Robani:

I'm pleased to inform you that your manuscript has been deemed suitable for publication in PLOS ONE. Congratulations! Your manuscript is now with our production department. 

Kind regards, 

on behalf of

Dr. Abdul Aziz Khan Niazi 

Academic Editor

PLOS ONE